# Physiological Responses and Histopathological Changes in Narrow-Clawed Crayfish (*Pontastacus leptodactylus*) Under Acute Thermal Stress

**DOI:** 10.3390/ani15131837

**Published:** 2025-06-21

**Authors:** Xia Zhu, Bin Li, Yuzhen Liu, Shujian Chen, Yangfang Ye, Ronghua Li, Weiwei Song, Changkao Mu, Chunlin Wang, Ce Shi

**Affiliations:** 1Key Laboratory of Aquacultural Biotechnology, Ningbo University, Chinese Ministry of Education, Ningbo 315211, China; zhuxia1027@126.com (X.Z.); liuyuzhen@nbu.edu.cn (Y.L.); yeyangfang@nbu.edu.cn (Y.Y.); lironghua@nbu.edu.cn (R.L.); songweiwei@nbu.edu.cn (W.S.); muchangkao@nbu.edu.cn (C.M.); wangchunlin@nbu.edu.cn (C.W.); shice3210@126.com (C.S.); 2Key Laboratory of Green Mariculture (Co-Construction by Ministry and Province), Ministry of Agriculture and Rural, Ningbo 315211, China; 3Collaborative Innovation Center for Zhejiang Marine High-Efficiency and Healthy Aquaculture, Ningbo 315832, China; 4College of Veterinary Medicine, Xinjiang Agricultural University, Urumqi 830052, China; libin19921221@126.com; 5Marine Economic Research Center, Donghai Academy, Ningbo University, Ningbo 315211, China

**Keywords:** thermal stress, energy metabolism, narrow-clawed crayfish, RNA-seq

## Abstract

Temperature significantly impacts aquatic organisms. This study comprehensively investigated the physiological responses and regulatory mechanisms of narrow-clawed crayfish (*Pontastacus leptodactylus*) under acute thermal stress, analyzing survival rate, metabolic response, histological characteristics, antioxidant capacity, and transcriptomic profiles. The findings demonstrate that the upper thermal tolerance threshold of this species is 25 °C, with the observed high mortality under elevated temperatures primarily attributable to three synergistic factors: heat-induced histopathological alterations in the hepatopancreas, dysregulation of energy metabolism homeostasis, and excessive accumulation of reactive oxygen species. These results provide crucial insights into thermal tolerance mechanisms in *P. leptodactylus* while offering scientific foundations for optimizing thermal stress management protocols and overcoming development obstacles of its north-south relay farming.

## 1. Introduction

The narrow-clawed crayfish, *Pontastacus leptodactylus* (Eschscholtz, 1823), is a freshwater species of significant ecological and economic value, widely distributed in Eastern Europe and the Middle East. Its global aquaculture production ranks fifth among farmed crayfish species [1,2]. In 2023, it was first reported in China’s Irtysh River Basin in Xinjiang, filling a gap in the natural distribution of crayfish in the region. Multiple studies have demonstrated that *P. leptodactylus* is a cold-water species whose reproductive season begins in autumn as water temperatures decline. Mating typically occurs between October and November at 7–12 °C, followed by egg-laying 4 to 6 weeks later at 6–11 °C. Its slow growth rate requires 2–3 years to reach marketable size [3,4,5,6]. Research further indicates that as a cold-water crayfish species, the maximum survival temperature for its cultivation is 25 °C or lower [7]. Since it reaches its thermal limit at 25 °C and grows better but has a lower survival rate at higher temperatures, the growth temperature of *P. leptodactylus* should be controlled between 20 and 25 °C to achieve the best yield [8,9]. However, the current low winter temperatures and prolonged ice-cover period in Xinjiang have made seasonal temperature variation a significant limiting factor for the local aquaculture of *P. leptodactylus.*

Against this backdrop, China’s “North-South relay farming” model offers a viable solution for *P. leptodactylus* aquaculture. “North-South relay farming” is a trans-regional collaborative aquaculture model grounded in geographical and climatic heterogeneity. Leveraging China’s pronounced latitudinal temperature gradients, this system implements phased migration of cultured species based on precise alignment with their thermal tolerance thresholds, thereby overcoming the environmental constraints of single-region cultivation. In the case of abalone farming, this model was first implemented in China in 2003, which involves transporting abalones from southern to northern sea areas from April to May each year to avoid summer high temperatures and then transporting them back to the south for overwintering until they mature in November [10,11]. This mode utilizes the temperature difference between the northern and southern areas, keeping animals in an appropriate water temperature environment throughout the farming period, improving their survival rate, shortening the farming cycle, and bringing greater economic benefits [12,13]. It can be seen that the technical core of the north-south relay farming model lies in precisely matching the temperature tolerance thresholds of the farmed species to achieve continuous production throughout the year. The North-South relay farming of *P. leptodactylus* could resolve production challenges caused by growth stagnation during ice-cover periods in Xinjiang’s northern regions, transforming a biennial production cycle into an annual one. As a cold-water-adapted species, *P. leptodactylus* enables year-round cultivation, offering a strategic advantage for seasonal market differentiation. This approach addresses the winter supply gap in southern regions and demonstrates substantial potential for expanding aquaculture sustainability and meeting escalating market demands. However, thermal stress remains a formidable challenge for the development of its North-South relay farming model.

Temperature, as an essential environmental factor for aquatic organisms, particularly poikilothermic animals like crustaceans, regulates their metabolism and critically influences their survival, growth, ontogeny, and reproduction [1,7,14]. Numerous studies have shown that crustaceans have a high survival rate and growth performance within an optimal temperature range [15]. For instance, for the juvenile Borelli’s prawn (*Macrobrachium borellii*) and Argentine freshwater prawn (*Palaemonetes argentinus*), the temperature range of 20–25 °C provides optimal conditions for achieving the highest survival rates and maximized growth rates [16]. In addition, the growth performance of red claw crayfish (*Cherax quadricarinatus*) and green shrimp (*Macrobrachium nipponense*) peaked at 24–28 °C and 22–32 °C, respectively, with significant declines observed beyond these ranges [17,18].

However, acute temperature shifts exceeding aquatic organisms’ tolerance threshold can trigger a series of physiological disruptions via diverse pathways, exerting direct effects on metabolic processes, chemical reaction rates, enzyme activity, and immune competence. It influences organisms across multiple levels of biological organization, spanning from the molecular to the whole-organism level [19,20,21,22]. Organisms maintain internal homeostasis and adapt to temperature fluctuations by altering their biochemical and metabolic processes. Under acute thermal stress, metabolic rates generally increase with rising temperatures in most aquatic organisms. Due to the stress response, energy is reallocated to homeostatic maintenance, resulting in a sharp increase in energy demand and disruption of energy metabolism [23,24]. Glucose (GLU), a critical metabolic fuel, is adjusted in accordance with the increase in energy demand. Similarly, lipid substances, such as triglycerides (TG) and total cholesterol (T-CHO), serve as crucial energy reserves under temperature stress, providing efficient support for maintaining energy supply. Crustaceans possess an open circulatory system (OCS) that relies on hemolymph for nutrient transport. Metabolites within the hemolymph serve as biomarkers of physiological adaptations to environmental shifts in these organisms [25,26]. As important indicators of stress response, GLU, TG, and T-CHO in hemolymph can reflect the changes in energy utilization under acute stress [27,28,29]. Metabolic processes inherently generate a certain quantity of reactive oxygen species (ROS). Additionally, acute temperature changes can induce excessive ROS accumulation, leading to oxidative stress and causing damage to cellular macromolecules such as DNA, proteins, and lipids [30,31]. Unsaturated fatty acids within biomembrane phospholipids are particularly vulnerable to ROS attack, resulting in lipid peroxidation and the accumulation of malondialdehyde (MDA), a key lipid peroxidation product [32]. To counteract oxidative stress, aquatic organisms possess an antioxidant system that includes antioxidant enzymes such as superoxide dismutase (SOD), catalase (CAT), and glutathione peroxidase (GPx) [33]. The hepatopancreas serves as a pivotal organ for nutrient metabolism and immunomodulation in crustaceans. This tissue harbors diverse enzymatic components associated with immune responses, collectively reflecting the functional status of the crustacean antioxidant defense system. Analysis of its histological architecture and molecular responses provides critical insights into thermal stress adaptation mechanisms in these organisms [34,35]. When the temperature exceeds the tolerance range that it has adjusted to, it will lead to the death of the individual. Thus, defining the temperature tolerance range of a species is an important prerequisite for its farming. And there is currently a lack of in-depth research on the physiological adaptability of *P. leptodactylus* under thermal stress.

Therefore, this study aims to elucidate the response and adaptation mechanisms of *P. leptodactylus* to acute thermal stress. We hypothesize that this species responds to thermal stress through metabolic pathway regulation, antioxidant system activation, and histopathological changes. These adaptations collectively determine survival outcomes. The findings are expected to advance the understanding of physiological adaptability and temperature tolerance in *P. leptodactylus*, drive research on adaptive domestication technologies, and promote the development of North-South relay farming for this species.

## 2. Materials and Methods

### 2.1. Ethics Statement

The research animals used in this study were not of species classified as endangered according to international conservation standards or of species protected by law. This experimental protocol was performed under the guidance of the Institutional Animal Care and Use Committee of Ningbo University (No. 20190410-042), ensuring strict compliance with contemporary regulations for the welfare of laboratory animals.

### 2.2. Experimental Animals and Rearing Conditions

*P. leptodactylus* used in the experiment with an initial body weight of 60.99 ± 6.93 g were wild-caught from the Irtysh River, Xinjiang Uygur Autonomous Region, China. Before the experiment, all crayfish were temporarily stocked in 6 tanks (Length × Width × Depth = 4 × 4 × 1 m), each stocked with 200 individuals in a 50 cm water column in the laboratory at Ningbo University. The crayfish were acclimated to the laboratory conditions (temperature: 15 ± 1 °C; dissolved oxygen: >7.5 mg/L; pH: 7.8 ± 0.4; total ammonia nitrogen < 0.1 mg/L; NO_2_-N < 0.1 mg/L) for 2 weeks. The freshwater was replaced every two days [1]. After 2 weeks, 320 healthy crayfish were selected (from the acclimated population) and randomly distributed into experimental tanks.

### 2.3. Experimental Design, and Sampling

In the thermal stress experiment, 4 different temperature treatments were set up, namely 15 °C (control group), 20 °C (T20), 25 °C (T25), and 30 °C (T30). Each treatment consisted of 4 replicate groups, with 20 crayfish per replicate (n = 4 replicates, 20 individuals/replicate, total 80 crayfish/treatment). The experimental narrow-clawed crayfish were transferred from the acclimation tank at 15 °C to preheated tanks maintained at 15 °C (control group), 20 °C, 25 °C, and 30 °C, respectively. Survival rates were recorded at 3 h, 6 h, 12 h, 24 h, 48 h, and 72 h after stress. At each time interval, 6 specimens of narrow-clawed crayfish were randomly selected from each treatment, and the hemolymph and hepatopancreas were collected (n = 6). The hemolymph samples were allowed to settle for 24 h at 4 °C before being centrifuged at 3000 rpm for 20 min, and the resulting supernatants were cryopreserved at −80 °C for biochemical assays. Hepatopancreas samples were immersion-fixed in 10% neutral buffered formalin (2 mL microtubes) for histological evaluation, while residual tissue was snap-frozen in liquid nitrogen and stored at −80 °C conditions for further analysis.

### 2.4. Histopathological Examination

For histological analysis, the hepatopancreas of each sampled crayfish underwent a 24-h fixation in 10% neutral buffered formalin (NBF). The fixative was subsequently washed off, and the samples were dehydrated in graded ethanol solutions before embedding. Thin sections (3–4 μm) were cut from the prepared wax blocks, stained with haematoxylin and eosin (HE), and sealed with neutral balsam [36,37,38]. Images of the sections were then observed and recorded under a light microscope (DN-800 M, NOVEL, Ningbo, China).

### 2.5. Analysis of Biochemistry and Antioxidant Capacity

To investigate the effects of acute heat stress on the biochemistry of the narrow-clawed crayfish, 5 energy metabolism-related indices in serum (triglyceride [TG], total cholesterol [T-CHO], glucose [GLU], high-density lipoprotein-cholesterol [HDL-C], and low-density lipoprotein-cholesterol [LDL-C]) were measured according to the manufacturer’s protocols by using commercially available kits (Jiancheng Bioengineering Institute, Nanjing, China).

To assess antioxidant capacity, hepatopancreas tissue extracts were obtained by homogenizing 0.1 g samples in ice-cold normal saline (0.9% NaCl solution cooled in 4 °C) at a 1:4 (*w*/*v*) ratio. Centrifugation of the homogenate was performed at 2500 rpm (600× *g*) for 15 min at 4 °C, and the supernatant was collected to determine the total antioxidant capacity (T-AOC), the activities of superoxide dismutase (SOD) and catalase (CAT), and malondialdehyde (MDA) content using commercial assay kits (Jiancheng Bioengineering Institute, Nanjing, China). Reactive oxygen species (ROS) levels were quantified with commercially available ELISA kits (Enzyme-linked Biotechnology, Qiaodu-Bio, Shanghai, China). All measurements were performed in triplicate (6 biological replicates with 3 technical replicates each) on the microplate reader (SpectraMax 190, Molecular Devices, San Jose, CA, USA).

### 2.6. RNA Extraction, Library Preparation, and Sequencing

To investigate the molecular mechanisms underlying the adaptation of narrow-clawed crayfish to acute thermal stress, hepatopancreas samples from each experimental group were subjected to total RNA extraction using the TRIzol Reagent Kit (Invitrogen, Carlsbad, CA, USA) following standard manufacturer protocols. RNA integrity was systematically evaluated through dual verification methods: quantitative analysis using an Agilent 2100 Bioanalyzer (Agilent Technologies, Palo Alto, CA, USA) and qualitative confirmation via RNase-free agarose gel electrophoresis.

Following total RNA extraction, mRNA was isolated using poly-T oligonucleotide-conjugated magnetic beads. RNA fragmentation was induced through divalent cation-mediated hydrolysis under elevated temperature in 5X First Strand Synthesis Reaction Buffer. First-strand cDNA synthesis was initiated with random hexamer primers and M-MuLV Reverse Transcriptase, followed by RNA template digestion via RNase H. Second-strand cDNA was generated using DNA Polymerase I and dNTPs, with residual overhangs converted to blunt ends through exonuclease/polymerase-mediated polishing. Adenylated 3′ termini were subsequently ligated to hairpin-structured adapters for hybridization readiness. Size selection (370–420 bp) was executed using the AMPure XP system (Beckman Coulter, Beverly, MA, USA). PCR amplification employed Phusion High-Fidelity DNA Polymerase with Universal PCR primers and Index (X) Primer, followed by AMPure XP purification. Final library quality was validated through dual-platform quantification (Qubit 2.0 Fluorometer) (INVITROGEN, Invelikij, USA) and electrophoretic profiling (Agilent Bioanalyzer 2100 system).

Index-multiplexed samples underwent automated cluster generation via the Illumina cBot system employing TruSeq PE Cluster Kit v3-cBot-HS (Illumina, San Diego, CA, USA), strictly adhering to established manufacturer protocols. Post-clustering, high-throughput sequencing was conducted on an Illumina Novaseq 6000 platform (Illumina, San Diego, CA, USA) utilizing a paired-end 150 bp (PE150) configuration, yielding dual-indexed sequence reads with 150 nucleotide phred quality scores at both termini.

### 2.7. De Novo Assembly, Unigene Annotation, and Enrichment Analysis

FASTQ format raw reads were quality-filtered using fastp (version 0.18.0), and the resulting clean reads were then used for *de novo* transcriptome assembly with Trinity (version 2.4). The longest transcript sequence from each gene was selected as the unigene after the assembly was completed.

Unigenes were functionally annotated via BLASTx (version 3.5.6.6) (E-value ≤ 1 × 10^−5^) against the following databases: NCBI non-redundant protein (Nr) database, the COG/KOG database, the Kyoto Encyclopedia of Genes and Genomes (KEGGs) database, and the Swiss-Prot protein database. Protein functional annotations were subsequently derived through computational analysis of optimal sequence alignment matches, utilizing highest-confidence BLAST hits for biological interpretation.

Expression data for the treatment and control libraries were generated by mapping reads to the transcriptome assembly using Bowtie2 (version 2.2.3). Gene estimation and quantification were then performed with RSEM (version 1.3.0). Unigene expression levels were calculated and normalized as RPKM (Reads Per kb per Million reads).

Finally, differentially expressed genes (DEGs) in the hepatopancreas between the control and two heat treatment groups (25 and 30 °C) were identified using the DESeq2 R package (version 1.44.0) [39]. A false discovery rate (FDR) was applied for multiple testing correction. Genes with fold-change values greater than 2 or less than 0.5 were considered as DEGs for further analysis. GO and KEGG pathway enrichment analyses were performed using the “ClusterProfiler” R package (version 4.2.2). GO terms with a corrected *p*-value less than 0.05 were deemed significantly enriched by DEGs. For KEGG enrichment, a *p*-value less than 0.05 and a *p.adjust* value less than 0.2 were used as criteria for significant enrichment.

### 2.8. Statistical Analysis

Survival rate (SR, %) was calculated as 100 × (final number of crayfish)/(initial number of crayfish). Data are expressed as the mean ± standard deviation (SD). Statistical analyses were performed using SPSS 22.0 (IBM SPSS Statistics, Chicago, IL, USA). Normality and homogeneity of variance were examined through the Kolmogorov–Smirnov test and the Levene test, respectively. Data meeting both assumptions were analyzed by one-way ANOVA, followed by Tukey’s *post hoc* test. Data for MDA (at 3, 12, and 24 h), TG (at 12 h), and T-CHO (at 3 h) did not meet the assumptions of normality and homogeneity of variances and were therefore analyzed using the non-parametric Kruskal–Wallis test followed by Dunn’s test. *p*-values less than 0.05 were considered statistically significant.

## 3. Results

### 3.1. Survival Rate

No significant effect on the survival rate occurred when the water temperature ranged from 15 to 25 °C (*p* > 0.05), and the survival rate remained at 100% during the stress time (72 h). Nevertheless, the T30 group showed a significantly decreased survival rate at 48 h compared to the control group (*p* < 0.05) (Figure 1).

### 3.2. Histological Changes in Hepatopancreas Following Acute Thermal Stress

The histological alterations of the hepatopancreas of the narrow-clawed crayfish after 48 h of acute thermal stress are depicted as follows. The hepatopancreas of the control group exhibited an intact tissue structure, with neatly arranged and plump epithelial cells that were readily distinguishable. Four primary cell types, namely (resorptive cells) R-cells, (fibrillar cells) F-cells, (blister-like cells) B-cells, and (embryonic cells) E-cells, could be clearly discerned; moreover, the star-shaped lumens were distinct, and the basement membranes were intact (Figure 2A,B). In crayfish exposed to 25 °C and 30 °C for 48 h, the hepatopancreas manifested histological damage to varying degrees. In the T25 group, the arrangement of hepatopancreas cells was slightly disordered, the number of B-cells increased significantly, they were hypertrophic and highly vacuolated, and the number of R-cells was lower compared to the control group (Figure 2C,D). In the T30 group, the arrangement of hepatopancreas cells was disorderly, cell necrosis was evident, and the boundaries between some cells vanished, cell fusion occurred, which made cell identification difficult (Figure 2F). Moreover, the condition of hepatopancreatic tubule dilation, lumen expansion, and disappearance of the star-shaped lumen structure was also observed in the T30 group; additionally, the hemocytes extravasated (Figure 2E).

### 3.3. Biochemical Parameters and Antioxidant Capacity

#### 3.3.1. Biochemical Parameters

In this study, the contents of TG, T-CHO, GLU, HDL-C, and LDL-C were investigated (Figure 3). After 6 h of thermal stress, TG content was significantly lower in the T25 and T30 groups compared to both the control and T20 groups (*p* < 0.05). At 48 h, however, the T20 group showed significantly higher TG levels than all other groups, peaking at 72 h (*p* < 0.05) (Figure 3A). Excluding the T20 group at the 72 h time point, the T-CHO content in each treatment group exhibited a significant reduction compared to the control group following 12 h of exposure (*p* < 0.05), and no significant differences were detected among the treatment groups (*p* > 0.05) (Figure 3B). Similarly, GLU content in all treatment groups significantly decreased compared to the control after 6 h of exposure (*p* < 0.05), and no significant differences were found among the treatment groups (*p* > 0.05) (Figure 3C). After 12 h of stress, HDL-C levels were significantly lower in all treatment groups compared to the control group (*p* < 0.05), but there were no significant alternations among the treatment groups at 72 h (*p* > 0.05) (Figure 3D). The LDL-C content in each treatment group also exhibited a descending tendency in contrast to the control group after 12 h of stress (*p* < 0.05), with no significant differences among treatment groups (*p* > 0.05) (Figure 3E).

#### 3.3.2. Antioxidant Capacity

ROS content under different temperature stresses is shown in Figure 4A. In crayfish exposed to 30 °C, the ROS content showed a significant increase compared with the other groups throughout the stress process (*p* < 0.05), and there were no significant alterations in ROS content between the control group and the T20 and T25 groups (*p* > 0.05). Compared with the control group, the T-AOC of each treatment group exhibited an ascending trend at each time point. Moreover, with the increase in temperature, T-AOC showed a trend of initial increase followed by a decrease. T-AOC content was significantly higher at 20 °C than in other groups (*p* < 0.05) (Figure 4B). Thermal stress affected hepatopancreatic SOD activity, with the T30 group showing a marked increase (peaking at 24 h) compared to other groups (*p* < 0.05). SOD activity in the T20 and T25 groups also significantly increased at 6 and 12 h relative to the control (*p* < 0.05), with no significant difference between the T20 and T25 groups at any time point (*p* = 0.105, >0.05) (Figure 4C). The CAT activity of the crayfish exposed to 30 °C for 3 h was elevated in comparison to the control group (*p* < 0.05). And there were no significant changes in CAT activity between each treatment group and the control group at 6 h (*p* > 0.05). Compared with the control group and T20 group, the CAT activity of the T25 and T30 groups increased after 12 h of stress, reaching peaks at 24 h and 12 h, respectively (*p* < 0.05). No significant difference in the CAT activity between the T20 group and the control group was detected throughout the stress process (*p* = 0.355, >0.05) (Figure 4D). Under 30 °C stress, the MDA content was conspicuously higher than the control group and T20 group at each time point (*p* < 0.05), reaching a peak at 3 h. At 6 h, 12 h, and 48 h, the MDA content of the crayfish exposed to 25 °C was significantly higher than the control group (*p* < 0.05) and reached a peak at 12 h. The T20 group did not show any significant variations compared with the control group during the entire stress process (*p* = 0.518, > 0.05) (Figure 4E).

### 3.4. Comparative Transcriptome Analysis

#### 3.4.1. Quantitative Analysis of DEGs in Hepatopancreas of the Narrow-Clawed Crayfish Under Acute Thermal Stress

Based on the above results, transcriptome analysis was performed on the hepatopancreas of the narrow-clawed crayfish treated at different temperatures at 48 h in order to investigate the response mechanism of crayfish to acute thermal stress. The results showed that in the hepatopancreas, 217 DEGs (140 up- and 77 down-regulated) in the control vs. T25 group and 4254 DEGs (4067 up- and 187 down-regulated) in the control vs. T30 group were identified (Figure 5A,B,D,E). In addition, compared with the T25 group, 1289 DEGs (1257 up- and 32 down-regulated) in the T30 group were identified (Figure 5C,F).

#### 3.4.2. Enrichment Analysis of DEGs in Hepatopancreas of the Narrow-Clawed Crayfish Under Acute Thermal Stress

To elucidate the biological functions of DEGs among different temperature treatment groups, GO and KEGG enrichment analyses were performed using ClusterProfiler in R (verison 4.5.0). GO enrichment results categorized DEGs into three ontologies: biological process (BP), cellular component (CC), and molecular function (MF). For the Control vs. T25 comparison, enriched BP terms included metabolic process, cellular process, response to stimulus, and biological regulation; enriched MF terms included catalytic activity, binding, transporter activity, and ATP-dependent activity; and enriched CC terms primarily involved cellular anatomical entity and protein-containing complex (Figure 6A). The Control vs. T30 comparison showed enrichment in BP terms such as metabolic process, cellular process, response to stimulus, biological regulation, and immune system process; enriched MF terms included binding, catalytic activity, transcription regulator activity, molecular function regulator, and ATP-dependent activity; and enriched CC terms primarily involved cellular anatomical entity and protein-containing complex (Figure 6B). Finally, the T25 vs. T30 comparison revealed significant enrichment in BP terms, including cellular process, metabolic process, and biological regulation; enriched MF terms included binding, catalytic activity, and transcription regulator activity; and enriched CC terms primarily involved cellular anatomical entity and protein-containing complex (Figure 6C).

KEGG enrichment analysis revealed that, for the control vs. T25 comparison, DEGs were significantly enriched in pathways including oxidative phosphorylation, metabolic pathways, carbon metabolism, and glycolysis/gluconeogenesis (*p* < 0.05) (Figure 7A). In the Control vs. T30 comparison, DEGs were significantly enriched in pathways such as spliceosome, nucleocytoplasmic transport, MAPK signaling pathway, FoxO signaling pathway, and apoptosis (*p* < 0.05) (Figure 7B). Finally, for the T25 vs. T30, the DEGs were significantly enriched in pathways such as spliceosome, protein processing in endoplasmic reticulum, MAPK signaling pathway, apoptosis, and glycolysis/gluconeogenesis (*p* < 0.05) (Figure 7C).

## 4. Discussion

As a primary abiotic factor, temperature profoundly affects the physiological mechanisms, survival, growth, metabolism, and immune responses of aquatic species, as demonstrated by numerous [40,41]. Therefore, when temperatures exceed or fall below the species-specific optimal range, individuals exhibit delayed growth, impaired reproductive performance, and increased mortality. In this study, an acute thermal stress gradient spanning 15–30 °C was established to investigate its effects on the survival rate, biochemical indices, hepatopancreatic histopathology, and antioxidant capacity of the *P. leptodactylus*. In addition, the molecular regulatory features underlying thermal adaptation were further elucidated using transcriptomic analysis. Notably, the survival rate in the T30 group decreased significantly after 48 h of exposure, whereas organisms at 15–25 °C maintained high survival rates, aligning with previous findings on the species’ upper thermal tolerance threshold [7,9] and confirming the narrow-clawed crayfish as a cold-water species.

### 4.1. How Does Thermal Stress Affect the Structure and Cellular Functions of the Hepatopancreas?

The results of this study demonstrated that the cellular morphology and structural organization of hepatopancreatic tissues were disrupted when reared under severe thermal stress (25 °C and 30 °C). In crustaceans, the hepatopancreas serves as a critical organ for metabolism, detoxification, and endocrine regulation [42,43]. Composed of hundreds of blind-end tubules and intertubular spaces, its fundamental functional unit is the hepatopancreatic tubule. The distal E-cells within these tubules generate three distinct epithelial cell types: B-cells for nutrient absorption and enzyme synthesis, R-cells for nutrient storage, and F-cells for structural support, all exhibiting pronounced sensitivity to temperature fluctuations [44,45,46]. In this study, B-cell density was significantly elevated in the T25 group, suggesting enhanced digestive and absorptive capacity to meet physiological demands under moderate thermal stress. Similar findings have been reported in *M. nipponense* [47], where high synthesis and release rates of digestive and antioxidant enzymes in B-cells accelerated nutrient mobilization within hepatopancreatic tubules, thereby providing additional energy to counteract environmental stressors [37]. Conversely, the reduction in R-cell populations may reflect increased metabolic consumption of nutrients under acute thermal stress, bypassing storage mechanisms [48]. Notably, the T30 group exhibited the most severe hepatopancreatic damage, characterized by disordered cellular arrangements, marked necrosis, and loss of morphological distinction. Hepatopancreatic tubules displayed pronounced dilation, lumen expansion, and disappearance of the star-shaped lumen structure, phenotypes consistent with observations in Kuruma shrimp (*Marsupenaeus japonicus*) under acute thermal stress [30]. Furthermore, thermal stress induced vacuolization of epithelial cells and lumen dilation in red swamp crayfish (*Procambarus clarkii*), impairing physiological functions and triggering irreversible tissue damage even after temperature normalization [49]. These findings indicate that extreme heat (30 °C) likely disrupts hepatopancreatic function. When thermal stress exceeds the hepatopancreas’s adaptive capacity, its core physiological functions—including energy metabolism and immune responses—are compromised, ultimately threatening organismal survival.

### 4.2. How Does Acute Thermal Stress Disrupt Energy Homeostasis and Metabolism in P. leptodactylus?

GLU, TG, and T-CHO, as core energy metabolism indicators, reflect the functional states of carbohydrate utilization, lipid storage, and transport systems, respectively [22]. Their dynamic variations directly mirror an organism’s energy supply capacity and homeostatic regulation under acute stress. In this study, the GLU levels in all treatments were significantly decreased compared to the control group after 6 h of thermal stress, with no significant differences observed among the temperature treatments. Similar results were also found in grass carp (*Ctenopharyngodon Idella*) [50]. To combat elevated temperatures, GLU is preferentially mobilized as a metabolic fuel, degraded through glycolysis and the tricarboxylic acid (TCA) cycle to generate ATP for rapid energy provision [51]. Acute thermal stress also profoundly impacts lipid metabolism. Results revealed significantly lower TG levels in the T25 and T30 groups compared to the control and T20 groups following 6 h of stress. By 12 h, T-CHO levels in all temperature-exposed groups declined markedly compared to controls. Similar results were reported in juvenile turbot (*Scophthalmus maximus*), where serum GLU, TG, and T-CHO levels decreased significantly under thermal stress [29]. Parallel observations in *C. quadricarinatus*, Rohu (*Labeo rohita*), and Nile tilapia (*Oreochromis niloticus*) further indicate that lipids are extensively catabolized to meet energy demands when glucose depletion fails to sustain metabolic requirements under intensified stress [52,53,54]. HDL-C and LDL-C levels reflect lipid decomposition, transport efficiency, and hepatopancreatic lipid metabolism [55]. In this study, HDL-C and LDL-C trends paralleled T-CHO dynamics, likely attributed to rapid cholesterol mobilization, suppressed hepatopancreatic function, and disrupted lipid metabolism under acute thermal stress. Notably, the T20 group exhibited significantly higher TG levels than other groups after 48 h of stress, peaking at 72 h. To maintain homeostasis under acute thermal stress, crayfish may accelerate lipid metabolism as a compensatory strategy when glucose-derived energy becomes insufficient, generating excess TG. However, metabolic substrates shift from anabolic to catabolic pathways due to either hyperactivated metabolic rates or reduced metabolic efficiency [23,24], explaining the elevated TG levels. Synchronized depletion of GLU, TG, and T-CHO correlated with markedly increased mortality, suggesting that alterations in energy metabolism may critically contribute to mortality rates.

### 4.3. How Does the Antioxidant System Navigate the Escalating ROS Burden Under Acute Thermal Stress?

The antioxidant system, encompassing enzymes such as SOD and CAT, constitutes a critical defense mechanism against oxidative stress in organisms [56,57]. Under normal physiological conditions, a dynamic equilibrium exists between ROS production and antioxidant capacity. However, persistent overaccumulation of ROS beyond the scavenging capacity of the antioxidant system ultimately leads to oxidative damage, with MDA serving as a key biomarker of lipid peroxidation [58]. These indicators collectively provide a robust assessment of the antioxidant status in crayfish. Previous studies have demonstrated that acute temperature stress significantly enhances ROS generation in aquatic organisms such as mud crab (*Scylla paramamosain*) and marine rotifer (*Brachionus plicatilis*) [59,60]. Consistent with these findings, the results of this study revealed a marked elevation in ROS levels throughout the stress period, indicating that high temperature (30 °C) induced mitochondrial electron transport chain dysfunction, thereby triggering rapid accumulation of ROS (particularly superoxide radicals [O_2_^−^] and hydrogen peroxide [H_2_O_2_]) in crayfish [61]. To counteract ROS overproduction, crayfish activated defense mechanisms by upregulating SOD and CAT activities. SOD activity in the T30 group peaked at 24 h, catalyzing the dismutation of O_2_^−^ into H_2_O_2_, while CAT subsequently detoxified H_2_O_2_ into non-toxic H_2_O and O_2_, thereby mitigating lipid peroxidation damage [62]. This coordinated response aligns with antioxidant strategies documented in white shrimp (*Litopenaeus vannamei*) and *S. paramamosain* [63,64]. Notably, when temperatures exceed species-specific tolerance thresholds, the antioxidant system becomes compromised, exacerbating oxidative injury. For instance, in *C. idellus*, exposure to 34 °C for 0.5 h resulted in significant reductions in SOD activity and T-AOC, alongside elevated MDA levels [65]. In this research, despite increased SOD and CAT activities and elevated MDA content, the enhanced antioxidant enzyme activities were insufficient to neutralize excessive ROS generated under acute thermal stress, leading to persistent tissue oxidative damage. T-AOC, reflecting systemic antioxidant capacity, exhibited an ascending trend across all treatment groups at each timepoint [62]. Furthermore, T-AOC displayed a biphasic temperature-dependent response: it initially increased but declined at extreme temperatures, with the T20 group showing significantly higher T-AOC levels than other groups. This suggests that the antioxidant system efficiently counteracts oxidative stress under moderate thermal challenges (20 °C) but becomes suppressed under severe stress. ROS accumulation directly correlates with mortality, and stress-induced alterations in energy metabolism further exacerbate survival outcomes. This phenomenon was substantiated by the aforementioned evidence of metabolic dysregulation.

### 4.4. What Transcriptomic Insights Unravel the Molecular Responses to Acute Thermal Stress in P. leptodactylus?

As depicted in Figure 5, acute thermal stress elicited a widespread upregulation of mRNAs in the hepatopancreas of *P. leptodactylus*. This extensive transcriptional phenomenon underscores a robust cellular and molecular stress response, reflecting *P. leptodactylus*’s concerted effort to activate various protective mechanisms against thermal challenges. These mechanisms likely include the increased synthesis of stress-response proteins (e.g., heat shock proteins, HSPs), antioxidant enzymes, and other factors crucial for maintaining cellular homeostasis under stressful conditions. Previous studies have indeed indicated a broad temperature tolerance range for *P. leptodactylus*, with reports suggesting its ability to withstand extreme temperatures from 4 °C to 32 °C [66]. Another study has shown that the mortality rate of *P. leptodactylus* was higher in the first four weeks under the 30 °C stress condition, but there was no mortality rate after the fourth week [7]. Such adaptability is likely extended to various physiological processes. For instance, just as the signal crayfish (*Pacifastacus leniusculus*) possesses a cold-active transglutaminase (TGase) that maintains high activity even at 4 °C [67], it is plausible that *P. leptodactylus* also relies on specific enzymes and biochemical pathways that operate efficiently at higher temperatures, contributing to its observed thermal resilience. While our current findings shed light on the immediate molecular responses, further investigations are warranted to fully elucidate *P. leptodactylus*’s long-term adaptability and recovery dynamics in warmer aquatic environments. The transcriptional profiling of hepatopancreas from *P. leptodactylus* also revealed pathways responding to acute thermal stress. GO enrichment analysis revealed that these DEGs in hepatopancreatic tissues were predominantly associated with molecular functions and biological processes. In the control vs. T25 group, DEGs were primarily enriched in metabolic processes, including ATP metabolic process and purine ribonucleoside triphosphate metabolic process, and catalytic activity (particularly oxidoreductase activity). In the control vs. T30 group, the DEGs mainly participated in metabolic processes and binding. These findings align with the experimental measurements of energy metabolism and oxidase activity presented in this article. KEGG pathway analysis demonstrated that the hepatopancreatic DEGs were principally involved in metabolic pathways and signaling pathways. In the control vs. T25 group, DEGs showed significant enrichment in multiple metabolic pathways, such as oxidative phosphorylation, metabolic pathways, carbon metabolism, and glycolysis/gluconeogenesis. As the metabolic center of crustaceans, the hepatopancreas participates in digestive enzyme production, lipid/carbohydrate metabolism, and organic compound biosynthesis/catabolism. These results demonstrate that thermal stress significantly affects its metabolic function and disrupts homeostasis, consistent with previous observations in *P. clarkii* [21], Chinese shrimp (*Fenneropenaeus chinensis*) [68], and *M. nipponense* [47]. Oxidative phosphorylation, a critical pathway for cellular energy production, is mechanistically linked to enhanced energy supply [69]. The genes involved in the energy metabolism were enriched, which indicated that the *P. leptodactylus* mobilized energy metabolism to counteract thermal stress through coordinated activation with associated metabolic pathways such as glycolysis/gluconeogenesis. Notably, significant enrichment of the ascorbate and aldarate metabolism pathway, a phenomenon previously documented in *P. clarkii*, was observed [70]. Ascorbic acid, a potent antioxidant, plays a pivotal role in neutralizing ROS generated under stress conditions. This metabolic adaptation suggests that under heat stress conditions, in addition to energy metabolism, cells may also protect the organism from oxidative damage by activating protective metabolic pathways. DEGs in control vs. T30 and T25 vs. T30 comparisons were predominantly enriched in signaling pathways and apoptosis-related processes. Key signaling pathways included MAPK, FoxO, ErbB, TGF-beta, Hippo, and Notch signaling pathways. TGF-beta pathway dysregulation correlates with autoimmune and inflammatory responses [71], while FoxO signaling is crucial in apoptosis, cell differentiation, DNA repair, and oxidative stress response and widely involved in the regulation of acute heat stress [72]. As downstream effector molecules of the TGF-beta signaling pathway, SMAD, particularly Smad3, plays a crucial role in TGF-beta-mediated transcriptional regulation and serves as an index of TGF-beta signaling activity. Research has consistently shown that Smad3 overexpression promotes TGF-beta-induced apoptosis [73], and Smad3 is also recognized as a target gene of the FoxO signaling pathway. In this research, the significant upregulation of Smad3, functioning as a common gene for both pathways, is implicated in enhancing the narrow-clawed crayfish’s resilience to acute thermal stress-induced damage. When the hepatopancreas fails to fulfill its normal physiological functions, it can trigger a range of pathological reactions, such as inflammation and immune deficiency. Previous studies have documented that elevated Smad3 expression can promote inflammatory cytokine production in hepatopancreatic tissue [73]. This observation corroborates the histopathological changes evident in the hepatopancreas at 30 °C in the current study. In addition, oxidative stress can also activate the MAPK signaling pathway, triggering AP-1 activation, initiating a cascading response that results in cell apoptosis. Increased AP-1 expression was induced by 30 °C stress exposure, as demonstrated in this study. Given its role as a crucial transcriptional regulator, AP-1 influences various cellular processes such as proliferation, differentiation, apoptosis, and inflammation [71]. The upregulation of AP-1 and Smad3 genes observed in this study suggests the triggering of inflammatory responses and apoptosis in the narrow-clawed crayfish under 30 °C stress. Caspase activity is a useful marker reflecting stress-induced apoptosis [74]. This study revealed temperature-induced apoptosis via both intrinsic (Cytc-Caspase9-Caspase3) and extrinsic (Caspase8-Caspase3) pathways, with elevated expression of Caspase9, Caspase8, Caspase3, and Cytc release observed in *P. leptodactylus*. Unlike *P. clarkii*, where apoptosis occurs solely through the extrinsic pathway [21], this biphasic apoptotic activation mechanism represents an adaptive strategy for ecological stress management. The enrichment of these pathways demonstrates that *P. leptodactylus* responds to oxidative damage, metabolic disorders, and inflammatory reactions induced by heat stress through multiple layers, including metabolic changes, antioxidant defense, and adaptive apoptosis.

## 5. Conclusions

This study demonstrated that the maximum tolerable temperature for this species is 25 °C, and the high mortality under elevated temperatures is potentially attributed to heat stress-induced structural damage in hepatopancreatic tissues, dysregulation of energy metabolism, and excessive accumulation of ROS. Crucially, the results highlight the necessity of avoiding aquaculture practices in water environments exceeding 25 °C for *P. leptodactylus*. These findings enhance the understanding of adaptive mechanisms underlying acute thermal stress responses in *P. leptodactylus* while providing theoretical guidance for refining the North-South Crayfish Relay industry. However, this study mainly focuses on acute thermal stress. Recovery experiments have not been conducted to assess whether the thermal damage is reversible. The long-term survival and adaptability of *P. leptodactylus* in warm waters still need to be studied and evaluated.

## Figures and Tables

**Figure 1 animals-15-01837-f001:**
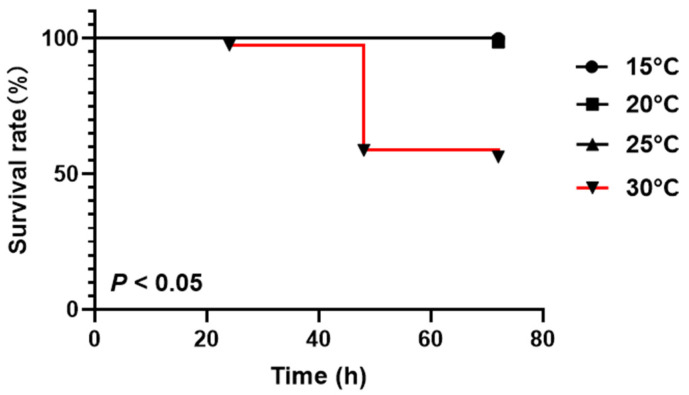
Survival rate of *P. leptodactylus* under acute thermal stress during a 72 h.

**Figure 2 animals-15-01837-f002:**
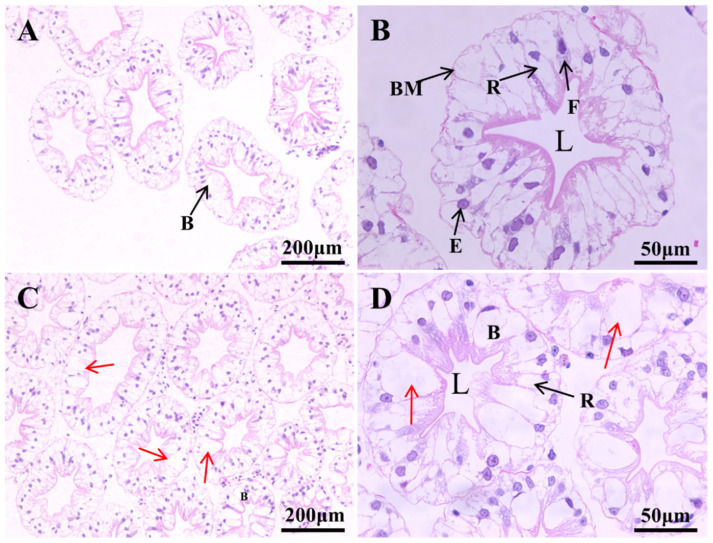
Transverse sections of the hepatopancreas tubules of the *P. leptodactylus* under acute high-temperature stress for 48 h. (**A**) 15 °C 100×, (**B**) 15 °C 400×, (**C**) 25 °C 100×, (**D**) 25 °C 400×, (**E**) 30 °C 100×, (**F**) 30 °C 400×; B: Blister-like cells, F: Fibrillar cells; R: Resorptive cells; E: Embryonic cells; L: Lumen; BM: Basement membrane. The red arrows indicate highly vacuolated cells, with some of the vacuoles being fused. The blue arrow indicates extravasation of hemocytes. The blue asterisk indicates the dilatation of the hepatopancreatic tubule lumen. The brown arrow indicates partial cell degeneration and fusion. The purple arrow points to damage in the brush border.

**Figure 3 animals-15-01837-f003:**
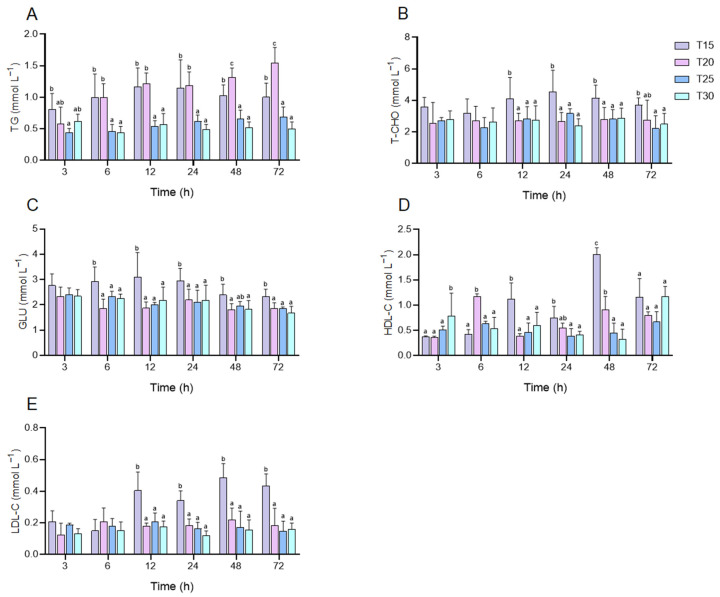
Changes in hemolymph content of TG (**A**), T-CHO (**B**), GLU (**C**), HDL-C (**D**), and LDL-C (**E**) in *P. leptodactylus* under thermal stress. Each bar represents the mean ± SD (n = 6). Different letters above the bars indicate that there are significant differences between groups (*p* < 0.05).

**Figure 4 animals-15-01837-f004:**
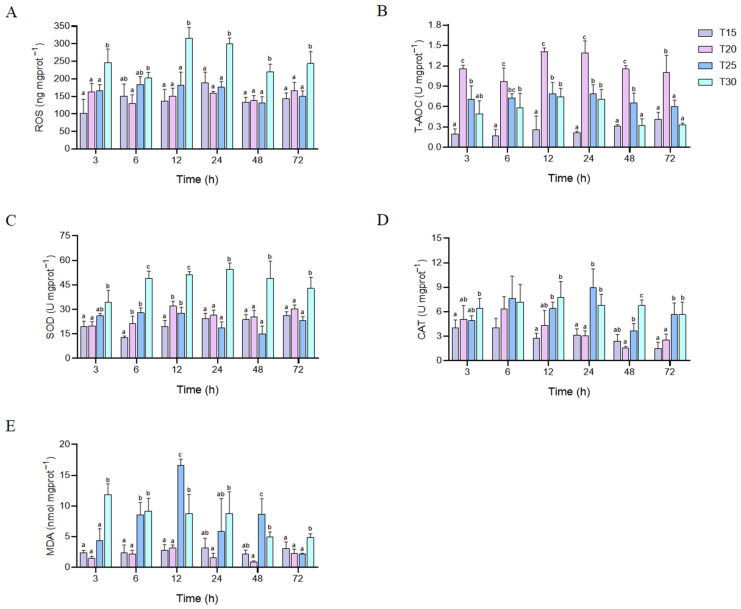
Changes in hepatopancreatic ROS content (**A**), T-AOC (**B**), SOD (**C**) and CAT (**D**) activities, and MDA content (**E**) in *P. leptodactylus* exposed to thermal stress. Each bar represents the mean ± SD (n = 6). Different letters above the bars indicate that there are significant differences between groups (*p* < 0.05).

**Figure 5 animals-15-01837-f005:**
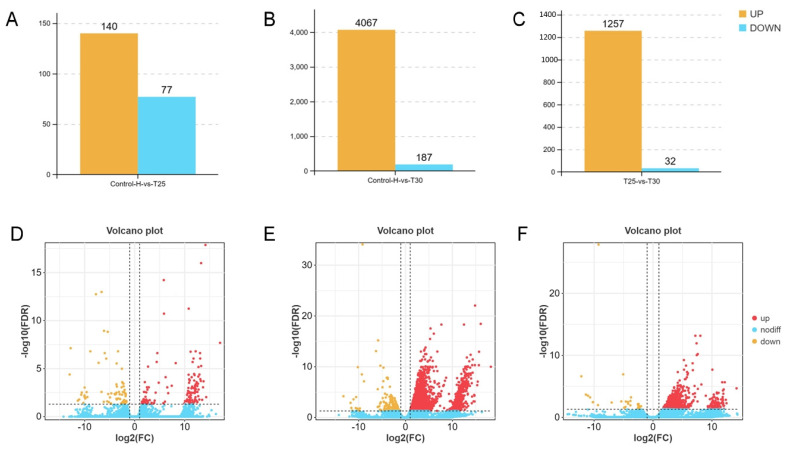
Differential gene expression analysis in the hepatopancreas of the *P. leptodactylus* under different temperature stress. Statistical analysis of DEGs of the Control vs. T25 (**A**), Control vs. T30 (**B**), and T25 vs. T30 (**C**). Volcano plot of DEGs of the Control vs. T25 (**D**), Control vs. T30 (**E**), and T25 vs. T30 (**F**).

**Figure 6 animals-15-01837-f006:**
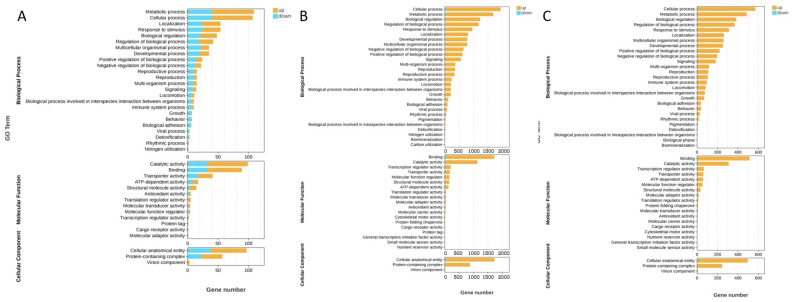
GO enrichment analysis of DEGs in the hepatopancreas of the *P. leptodactylus* under different temperature stress. Control vs. T25 (**A**), Control vs. T30 (**B**), and T25 vs. T30 (**C**).

**Figure 7 animals-15-01837-f007:**
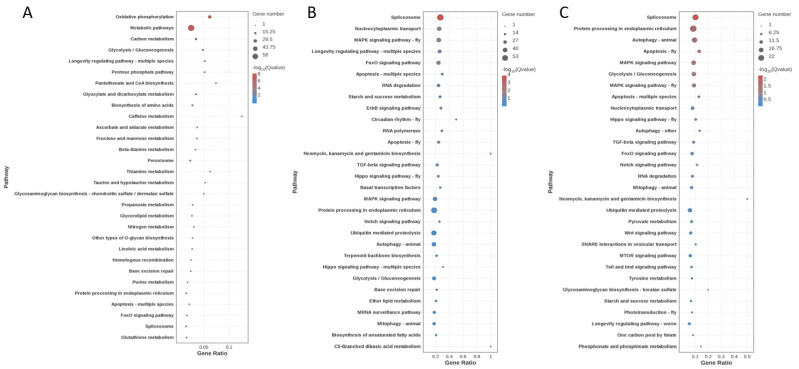
KEGG enrichment analysis of DEGs in the hepatopancreas of the *P. leptodactylus* under different temperature stress. Control vs. T25 (**A**), Control vs. T30 (**B**), and T25 vs. T30 (**C**).

## Data Availability

All data presented in this study are available on reasonable request from the corresponding author.

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
