# Peer review of "Physiological Responses and Histopathological Changes in Narrow-Clawed Crayfish (Pontastacus leptodactylus) Under Acute Thermal Stress"

_animals, 2025, doi:10.3390/ani15131837_

Round 1

Reviewer 1 Report

Comments and Suggestions for Authors

This manuscript, 'Physiological responses and histopathological changes to acute thermal stress in the narrow-clawed crayfish (Astacus leptodactylus)', showcases outstanding research on the thermal stress responses in the narrow-clawed crayfish. 

The manuscript demonstrates exceptional writing and presentation quality, featuring a clear and logical structure that facilitates effortless navigation. However, to further enhance the manuscript's quality, I have identified several issues, incorporated in PDF, that necessitate major revisions.

Author Response

Reviewer #1:

This manuscript, 'Physiological responses and histopathological changes to acute thermal stress in the narrow-clawed crayfish (Astacus leptodactylus)', showcases outstanding research on the thermal stress responses in the narrow-clawed crayfish. 

The manuscript demonstrates exceptional writing and presentation quality, featuring a clear and logical structure that facilitates effortless navigation. However, to further enhance the manuscript's quality, I have identified several issues, incorporated in PDF, that necessitate major revisions.

Authors response: We sincerely thanks for your positive assessment of our work and your recognition of the manuscript's scientific value and clarity! The encouraging comments are greatly appreciated and reinforce our commitment to rigorously communicating this research. We have carefully addressed all points raised in the annotated PDF. The manuscript has been comprehensively revised to incorporate the suggested improvements. For detailed answers, please refer to the Word document.

Comments 1: Temperature exerts important effects on aquatic organisms. Through comprehensive analyses of survival rate, metabolic response, histological characteristics, antioxidant capacity, and transcriptomic profiles, this study elucidates the physiological response characteristics and regulatory mechanisms of narrow-clawed crayfish (Astacus leptodactylus) under acute thermal stress.

Change to:Temperature significantly impacts aquatic organisms. This study comprehensively investigated the physiological responses and regulatory mechanisms of narrow-clawed crayfish (Astacus leptodactylus) under acute thermal stress, analyzing survival rate, metabolic response, histological characteristics, antioxidant capacity, and transcriptomic profiles.

Authors response: Thanks for the comment. We have made the necessary revisions to the text according to your suggestions. In addition, we have changed the scientific name of the narrow-clawed crayfish to the now more commonly used Pontastacus leptodactylus, although the Astacus leptodactylus is also a correct scientific name, as described in lines 24-30. We once again express our gratitude for your valuable suggestions that have helped further improve the paper!

Comments 2: In this study :Lack of detailed methodology

Authors response: Thanks for the comment. We have already refined the description of the method in the summary section, as described in lines 44-51.

Comments 3: Introduction:

-There are some minor grammatical errors that should be corrected

-Some transitions between paragraphs could be smoother.

Authors response: Thanks for the comment. We checked and corrected all the grammatical errors in the entire introduction section, and rearranged the overall paragraph order of the introduction to make it read more smoothly and fluently. We drew inspiration from the writing style of one of the recommended papers and added the advantages of using the hepatopancreas in the introduction section.

Comments 4: Histopathological examination:Which protocol did the authors follow?

-https://doi.org/10.1007/s11270-024-07643-y

-https://doi.org/10.1007/s12562-021-01511-y

Authors response: Thanks for the comment. We adopted this approach -(https://doi.org/10.1016/j.scitotenv.2019.02.159, https://doi.org/10.1016/j.aquaculture.2014.01.029). Similar to the one you provided (https://doi.org/10.1007/s11270-024-07643-y). We have already cited these articles in the chapter on histopathological examination, as described in line 266.

Comments 5: Biochemical parameters and antioxidant capacity:slightly concise the results of biochemical and antioxidant capacity

Authors response: Thanks for the comment. We realized that the expression of this part was indeed a bit lengthy, and we have made appropriate modifications to it, as described in lines 429-431, 471-475.

Comments 6: Discussion:Correct the sentence structure errors in whole discussion section.

Authors response: Thanks for the comment. We conducted a thorough examination of the sentence structures in the entire "discussion" section and also incorporated the suggestions you provided.

Comments 7: As a primary abiotic factor, temperature, as numerous studies have demonstrated, governs and constrains the physiological mechanisms, and profoundly affects the survival, growth, metabolism, and immune responses of aquatic species.

Change to:As a primary abiotic factor, temperature profoundly affects the physiological mechanisms, survival, growth, metabolism, and immune responses of aquatic species, as demonstrated by numerous studies.

Authors response: Thanks for the comment. We have made corresponding revisions in the "Discussion" section, as described in lines 562-565.

Comments 8: in contrast, organisms maintained at 15-25°C exhibited sustained high survival rates. This observation aligns with prior findings on the species’ upper thermal tolerance threshold [3, 35], which confirms the narrow-clawed crayfish as a cold-water species.

Change to:whereas organisms at 15-25°C maintained high survival rates, aligning with previous findings on the species' upper thermal tolerance threshold [3, 35] and confirming the narrow-clawed crayfish as a cold-water species.

Authors response: Thanks for the comment. We have made corresponding revisions in the "Discussion" section, as described in line 575-579.

Comments 9: Conclusions:-The manuscript requires revision to address grammatical, sentence structure, and spacing issues.

Authors response: Thanks for the comment. We have made revisions to the summary and added an outlook for this experiment. We once again express our gratitude for your constructive suggestions, which have significantly enhanced the quality of this work.

Reviewer 2 Report

Comments and Suggestions for Authors

Title : Physiological responses and histopathological changes to acute thermal stress in the narrow-clawed crayfish (Astacus leptodactylus)

I found the paper to be innovative and captivating; therefore, I support its publication in the journal after a slight revision. Below are my thorough observations and suggestions.

Specific recommendations :

-lines 2-4: Could you please change the title to the interrogative form; it will be more attractive for the readers. No experimental study give confirmed results at 100 %.

-Lines 141-147: Authors are invited to give hypotheses at the end of the introduction. Then, in their conclusion section, they must respond.

-Lines 160-161: Give a reference, please, for two days. Also, are you sure that no stress variation will occur during that period? Thereafter, how to be sure that no significant variation of stress magnitude will occur?

Are you giving food? If yes, what exactly?

-The section Materials and Methods is with no reference!!

-Lines 254-255: You have a low number of animals and replicates. So, two options: to use Kruskal-Wallis and Dunn test (non-parametric way) ot to transform to reach normality – Which option authors use, especially that the data are in %, probably the most appropriate transformation for ANOVA will be square root.

-Lines 327, 332, and 338: Please indicate the exact value of probability.

-Figure 4: for ROS, please check again the significance level for G30 after 12 and 24h .. seems it will be ‘c’, not ‘b’

Please indicate also if you used raw data or transformed data.

-Figures 6 and 7: please enhance the resolution, could not seen data, too small.

Discussion: Could you please transform the subtitles into questions?

Please avoid using ‘our’ throughout the main text. Consider instead the passive voice.

Line 397: ‘suppressed under severe stress’ do you mean inactive or inefficient here?

Conclusions: Please speak a little bit about the limitations of the work. In fact, without using microplastics well identified, it is a little bit difficult to generalize results. This section must be related clearly to the hypotheses and questions posed at the end of the introduction.

-References: check it again; there are some mistakes (journal name in full and not in abbreviation, commas, …). Eg Lines 618-619. Also, there is a strange dominance of Asian authors!

Author Response

Title : Physiological responses and histopathological changes to acute thermal stress in the narrow-clawed crayfish (Astacus leptodactylus).I found the paper to be innovative and captivating; therefore, I support its publication in the journal after a slight revision. Below are my thorough observations and suggestions.

Authors response: We sincerely thanks for your recognition of our work's innovation and your supportive recommendation for publication. The constructive feedback that you provided has been invaluable in refining this manuscript. We have meticulously addressed all observations and suggestions point-by-point as detailed below.

Comments 1: -lines 2-4: Could you please change the title to the interrogative form; it will be more attractive for the readers. No experimental study give confirmed results at 100 %.

Authors response: Thanks for the comment. We agree with your point of view and have already changed the title to an interrogative format in accordance with your suggestion.

Comments 2: -Lines 141-147: Authors are invited to give hypotheses at the end of the introduction. Then, in their conclusion section, they must respond.

Authors response: Thanks for the comment. We have already put forward the hypothesis at the end of the introduction, and it also corresponds with the conclusion, as described in lines 208-211.

Comments 3: -Lines 160-161: Give a reference, please, for two days. Also, are you sure that no stress variation will occur during that period? Thereafter, how to be sure that no significant variation of stress magnitude will occur?

Are you giving food? If yes, what exactly?

Authors response: Thanks for the comment. We referred to this piece of literature for two days: Alvanou, M. V.; Feidantsis, K.; Lattos, A.; Stoforiadi, A.; Apostolidis, A. P.; Michaelidis, B.; Giantsis, I. A., Influence of temperature on embryonic development of Pontastacus leptodactylus freshwater crayfish, and characterization of growth and  

osmoregulation related genes. BMC Zoology 2024, 9, (1), 8. https://doi.org/10.1186/s40850-024-00198-9. We have already cited it in the manuscript, as described in line 236. Apart from this article, many other studies that carry out water changes during the rearing period even the stress period. The frequency of water change varies from once a day to once every three days, in order to ensure that the crayfish can survive in good water quality. DOI: -10.18869/acadpub.ijaah.2.1.45; 10.1016/j.cbpc.2023.109581; 10.1016/j.fsi.2020.03.017; 10.1016/j.ecolind.2018.11.044. During the entire adaptation period, the crayfish exhibited the normal behaviors of the species and remained vigorous without any deaths. We will replace two-thirds of the fresh water to ensure that the crayfish do not suffer from oxygen deficiency (500L with 200 crayfish), under this density and water body condition, the water quality parameters will not cause stress to the animals. After the two-week adaptation period is over, we will select the high-quality and healthy crayfish for the subsequent stress experiments.

During the experiment, we did not provide any food.

Comments 4: -The section Materials and Methods is with no reference!!

Authors response: Thanks for the comment. We have included the relevant references throughout the entire Materials and Methods section to substantiate the methodological rigor of our approach.

Comments 5: -Lines 254-255: You have a low number of animals and replicates. So, two options: to use Kruskal-Wallis and Dunn test (non-parametric way) to transform to reach normality – Which option authors use, especially that the data are in %, probably the most appropriate transformation for ANOVA will be square root.

Authors response: Thanks for the comment. Before the analysis, our original data were all tested to determine whether they followed a normal distribution and whether the variances were homogeneous. Based on the tested data, one-way ANOVA and Tukey's multiple comparison post-hoc test were conducted to determine whether there were significant differences among the various treatments. For the data that did not meet the homogeneity analysis, we used the Kruskal-Wallis and Dunn tests (non-parametric methods), and this has been added in lines 367-376.

Comments 6: -Lines 327, 332, and 338: Please indicate the exact value of probability.

Authors response: Thanks for the comment. We have already marked the specific value of p in this part, as described in lines 471, 480, 487.

Comments 7: -Figure 4: for ROS, please check again the significance level for G30 after 12 and 24h .. seems it will be ‘c’, not ‘b’

Please indicate also if you used raw data or transformed data.

Authors response: Thanks for the comment. We rechecked the significance of the ROS once again. What we did was to compare the temperature groups at each time point. The significance level for the G30 group after 12 hours and 24 hours of exposure was indeed ‘b’. And we used mean ± SD of the raw data.

Comments 8: -Figures 6 and 7: please enhance the resolution, could not seen data, too small.

Authors response: Thanks for the comment. We have adjusted the resolution of Figures 6 and 7 to make them clearer. Now the data in the Figures can be seen.

Comments 9: Discussion: Could you please transform the subtitles into questions?

Authors response: Thanks for the comment. We have already set up question-format subtitles in the discussion section, as described in lines 580-581, 621-622, 665-666, 716-717.

Comments 10: Please avoid using ‘our’ throughout the main text. Consider instead the passive voice.

Authors response: Thanks for the comment. We have thoroughly reviewed the entire "discussion" section and have changed the original "our" to the passive voice.

Comments 11: Line 397: ‘suppressed under severe stress’ do you mean inactive or inefficient here?

Authors response: Thanks for the comment. The term "suppressed" specifically denotes reduced functional efficiency (not complete inactivation) of the antioxidant system under severe thermal stress. The activities of SOD and CAT remain relatively high at 30 degrees, but the content of MDA has also increased. This indicates that the antioxidant capacity cannot completely neutralize the reactive oxygen species, but it does not mean that the system has completely lost function.

Comments 12: Conclusions: Please speak a little bit about the limitations of the work. In fact, without using microplastics well identified, it is a little bit difficult to generalize results. This section must be related clearly to the hypotheses and questions posed at the end of the introduction. 

Authors response: Thanks for the comment. We have included a description of the limitations of this study in the conclusion section, as described in lines 843-848. And we have explicitly mapped the conclusion to the hypotheses and questions posed in the Introduction.

Comments 13: -References: check it again; there are some mistakes (journal name in full and not in abbreviation, commas, …). Eg Lines 618-619. Also, there is a strange dominance of Asian authors!

Authors response: Thanks for the comment. We sincerely apologize for the formatting oversights. We have conducted a thorough review and revision of the references. Crustacean thermal physiology is intensely studied in Asia due to: Regional Aquaculture Significance: Asia produces >85% of global farmed crustaceans (FAO, 2022). When citing literature, we give priority to considering its scientific relevance. And in this revision, we have also incorporated many research results from regions other than Asia. We once again sincerely thank you for your valuable contribution to ensuring the rigor and comprehensiveness of the paper.

Reviewer 3 Report

Comments and Suggestions for Authors

This study aimed to elucidate the response and adaptation mechanisms of A. leptodactylus to acute thermal stress by analyzing its survival rate, metabolism, antioxidant activity, tissue damage, and comparative transcriptome profiles. It is a very simplistic study but can be of some interest if the authors revise the manuscript. Most of the results are showing minor effects but still they have some interest but the Figures 6 and 7 need to be made larger since as it is now they are too small to read properly. In addition there may be a need for more animals to make the data and conclusion more robust.

All enzymatic analysis were made with kits and so how do we know that they wre made under correct kinetic conditions?

What is ”ice cold normal saline”?

How much g is 2500 rpm?

How many animals were used for the experimnets in Figure 1 for each temperature?

In Figure 2 the authors use German names for cells (zellen) so it seems as they do not know what these cells are? Blazenzellen what are they? Fibillinezelln what are they in English? This seems for this reviewer extremely strange to use Germna names  for cells which have an English name as well so revise please! Are the changes observed significant?

In Figure 3 the experiments do now show any significant differences  so the authors must increase the numberof animals and exposure times to reveal whether there are any effects?

In Figure 3 and 4 were samples during the incubation period taken from the same individual or was each sample taken form different individuals. This is very important to know if you are to intrepret data correct?

In Figure 5 it appears as if a majoriy of mRNAs are upregulated if exposd to stress! What does this mean for the animals after exposure? Were any experimnets done to reval how long time they required to adapt to normal conditions? In other words can this cold-adapted species live in warm water.

It must be of interest to read and cite a study in which it is shown that a cold-adapted crayfish has a cold-adapted enzyme namely transglutaminase. Sirikharin,R, Söderhäll,K and Söderhäll,I. 2018. Characterization of a cold-active transglutaminase from the crayfish, Pacifasacus leniusculus. Fish Shellfish Immunology 80,546-549 doi: 10.1016/j.fsi.2018.06.042. Epub 2018 Jun 27.PMID: 29960064

Comments on the Quality of English Language

Can be improved especially to sue English instead of German for cells in a tissue!!

Author Response

This study aimed to elucidate the response and adaptation mechanisms of A. leptodactylus to acute thermal stress by analyzing its survival rate, metabolism, antioxidant activity, tissue damage, and comparative transcriptome profiles. It is a very simplistic study but can be of some interest if the authors revise the manuscript. Most of the results are showing minor effects but still they have some interest but the Figures 6 and 7 need to be made larger since as it is now they are too small to read properly. In addition there may be a need for more animals to make the data and conclusion more robust.

Authors response: Thank you for your constructive feedback on our manuscript. We appreciate your recognition of the study’s potential interest and have carefully addressed all comments, revising the manuscript accordingly. We appreciate the reviewer's candid assessment. We acknowledge that the current study primarily focuses on the acute responses of P. leptodactylus to thermal stress, serving as an initial exploration of its coping mechanisms. We believe the insights gained from integrating physiological and transcriptomic data provide a valuable foundation for future, more comprehensive investigations into the long-term adaptation of this species. We have also expanded our discussion to better contextualize our findings within the broader understanding of cold-adapted species' responses to thermal challenges, highlighting both stress responses and potential adaptive plasticity.

We have regenerated Figures 6 and 7 at higher resolution.

We acknowledge the reviewer’s concern for the sample size and provide the following clarifications: While we understand the suggestion for more animals, the present study's findings, particularly the consistent trends observed across multiple physiological and molecular endpoints, provide a coherent picture of the acute thermal stress response in P. leptodactylus. And our data has passed the statistical normal distribution test. We are confident that the current sample size, combined with rigorous statistical analysis, adequately supports the conclusions drawn for the acute effects examined. We acknowledge this as a valuable point for consideration in future experimental designs, especially when investigating more subtle or long-term effects.

Comments 1: All enzymatic analysis were made with kits and so how do we know that they wre made under correct kinetic conditions?

Authors response: Thanks for the comment. We sincerely thank the reviewer for this insightful and critical question regarding the kinetic conditions of our enzymatic assays. We agree that ensuring optimal assay conditions is crucial for accurate enzyme activity measurements. We chose these commercially available assay kits primarily for their standardization, high reproducibility, and established methodologies, which are widely accepted and employed in biochemical and physiological research. In conducting these assays, we strictly adhered to the manufacturer's recommended protocols. These protocols are typically optimized to provide reliable measurements across a broad range of biological samples and are based on extensive biochemical characterization. We conducted dilution pre-experiments on both the serum and the hepatopancreas to ensure that the measured data and reading times were all within the linear range of the reagents. While these standardized conditions may not represent the absolute optimal kinetic conditions for P. leptodactylus enzymes under all possible stress scenarios, they allow for consistent and reproducible comparisons of enzyme activities between different experimental groups (control vs. stress-treated). The clear and consistent trends observed in our antioxidant enzyme activities, in conjunction with other physiological and molecular indicators (e.g., MDA content, transcriptomic data), provide a comprehensive picture of the antioxidant response to acute thermal stress.

Comments 2: What is “ice-cold normal saline”?

Authors response: Thanks for the comment. “ice-cold normal saline” refers to a 0.9% NaCl solution that has been cooled to 4°C. We followed this paper:https://doi.org/10.1016/j.aquaculture.2022.739080.

Comments 3: How much g is 2500 rpm?

Authors response: Thanks for the comment. The centrifugal force was calculated as 600 g using a rotor radius of 8.6 cm. The conversion formula was: RCF=1.118×10−5×8.6×(2500)2.

Comments 4: How many animals were used for the experimnets in Figure 1 for each temperature?

Authors response: Thanks for the comment. There are 80 crayfish at each temperature treatment group in Figure 1 (n = 4 replicates, 20 individuals/replicate, total 80 crayfish/treatment).

Comments 5: In Figure 2 the authors use German names for cells (zellen) so it seems as they do not know what these cells are? Blazenzellen what are they? Fibillinezelln what are they in English? This seems for this reviewer extremely strange to use Germna names for cells which have an English name as well so revise please! Are the changes observed significant?

Authors response: Thanks for the comment. We are sorry that we failed to notice this. We selected a reference article which uses this particular format. DOI: 10.1016/j.ecolind.2018.11.044. We mistakenly believed that this form was also acceptable. We know the full names of these cells are blister-like cells, fibrillar cells, resorptive cells, and embryonic cells. We also know the specific functions of these cells. We feel sorry that our reference was not comprehensive. We have made the necessary revisions in the manuscript, as described in lines 394-395, 417-418. The histopathological changes were significant.

Comments 6: In Figure 3 the experiments do now show any significant differences so the authors must increase the number of animals and exposure times to reveal whether there are any effects?

Authors response: Thanks for your comments. In figure 3, the contents of TG, T-CHO, GLU, HDL-C, and LDL-C in hemolymph were tested among treatments, with 6 independent individuals employed from each treatment. As such, this sample size meets the requirement of more than 3 biological replicates, which is sufficient to satisfy statistical needs and comply with publication norms of academic journals. In addition, in figure 3, all tested indicators showed significant differences, which indicating that the thermal stress obviously impacts the energy metabolism.

Comments 7: In Figure 3 and 4 were samples during the incubation period taken from the same individual or was each sample taken form different individuals. This is very important to know if you are to intrepret data correct?

Authors response: Thanks for the comment. Each sample was collected from different individuals (6 biological replicates with 3 replicates each treatment).

Comments 8: In Figure 5 it appears as if a majoriy of mRNAs are upregulated if exposd to stress! What does this mean for the animals after exposure? Were any experimnets done to reval how long time they required to adapt to normal conditions? In other words can this cold-adapted species live in warm water.

Authors response: Thanks for your constructive comment. This transcriptional phenomenon underscores a robust cellular and molecular stress response, reflecting P. leptodactylus's concerted effort to activate various protective mechanisms against thermal challenges. Previous studies have indeed indicated a broad temperature tolerance range for P. leptodactylus, with reports suggesting its ability to withstand extreme temperatures from 4°C to 32°C. Another study has shown that the mortality rate of P. leptodactylus was higher in the first four weeks under the 30°C stress condition, but there was no mortality rate after the fourth week. Such adaptability is likely extended to various physiological processes. While our current findings shed light on the immediate molecular responses, further investigations are warranted to fully elucidate P. leptodactylus's long-term adaptability and recovery dynamics in warmer aquatic environments. We have combined this part with the paper you provided and placed it in the discussion section, in lines 718-744.

Comments 9: It must be of interest to read and cite a study in which it is shown that a cold-adapted crayfish has a cold-adapted enzyme namely transglutaminase. Sirikharin,R, Söderhäll,K and Söderhäll,I. 2018. Characterization of a cold-active transglutaminase from the crayfish, Pacifasacus leniusculus. Fish Shellfish Immunology 80,546-549 doi: 10.1016/j.fsi.2018.06.042. Epub 2018 Jun 27.PMID: 29960064

Authors response: Thanks for your constructive comment. We have already cited this paper in the discussion section, in line 737. And we combined this with the issue you raised earlier regarding the upregulation of mRNA and conducted a discussion, in lines 718-744.

Reviewer 4 Report

Comments and Suggestions for Authors

I read the article “Physiological responses and histopathological changes to acute thermal stress in the narrow-clawed crayfish (Astacus leptodactylus)”. I would like to note the high level of work done. The experiment is well designed and includes a large number of methods for determining exposure. The text of the article is properly structured and understandable. I have only a few comments that I think would help to improve the manuscript.

Design comments

I suggest using the full names of biological species (with the year and author of the description) when they are first mentioned.

Usually in MDPI journals the reference to a figure in the text is written in full, not Fig.

No DOI of articles in the reference list

Please note that equipment should be fully cited with the model, company, city and country of manufacture. (For example, lines 182, 198.)

Line 244. Reference Love et al., 2014 - should be replaced by a serial number. In addition, it seems that this reference is not in the reference list.

It is not clear why it is necessary to highlight chapter 6 Patents in a separate section.

Simple summary: In my opinion, the phrase “development of its industry” is too generalized in this context. It is more appropriate to specify the need for research for aquaculture.

Introduction

lines 61-63. It is not clear why the qualifier “especially for aquatic animals” is used if it says aquatic organisms. Who do you mean in the first and second cases?

Lines 73-76. This sentence needs to be rephrased The verb “Rely” is misleading.

In general for this section, it seems to me more logical to describe the species being studied first, and then move on to the relevance of the study to describe the species being studied.

In addition, it would be useful in this section to justify the advantages of using hemolymph cells and hepatopancreas tissues as “targets” over other organs and cells.

Methodology

lines 157-160. What number of days the individuals were acclimated before the experiment. What were the hydrochemical parameters of the water?

Lines 163-165. Please clarify: if there were 4 groups of 4 replicates, and 20 crayfish in each group, the total number of crayfish is 320 individuals, while line 159 indicates 200. Or were there 6 tanks with 200 crayfish each? Then it is not clear why so many individuals were acclimated.

Then lines 168-170. If 6 individuals were randomly selected at each time point from 4 replicates, it appears that 1 or 2 individuals were taken from each replicate. What does this have to do with it?

Results

Lines 260-262 Two sentences contradict each other. The first sentence can be removed without loss of meaning. Why is the significance level 0.05 in the text and 0.0001 in Figure 1?

Figures 3 and 4. Why are the temperature points labeled T15 (20...) in the text and G15 in the figure? Indicate what the letters a,b,c stand for. Why is it stated (n = 3 replicates) when the methods say 4 replicates.

Author Response

I read the article “Physiological responses and histopathological changes to acute thermal stress in the narrow-clawed crayfish (Astacus leptodactylus)”. I would like to note the high level of work done. The experiment is well designed and includes a large number of methods for determining exposure. The text of the article is properly structured and understandable. I have only a few comments that I think would help to improve the manuscript.

Authors response: Thank you for your thoughtful review of our manuscript. We sincerely appreciate your positive feedback regarding the experimental design, methodological scope, and clarity of the manuscript. We are grateful for your recognition of the work’s rigor and structure. We appreciate your constructive suggestions and are committed to thoroughly addressing each of your proposals in order to enhance the quality of the manuscript. Below is a detailed response to your specific suggestions, along with the modifications made to the paper.

Comments 1: I suggest using the full names of biological species (with the year and author of the description) when they are first mentioned.

Authors response: Thanks for the comment. We have added "year and author" on line 86. Also, the original name "Astacus leptodactylus" has been renamed to the genus "Pontastacus", so we use "Pontastacus leptodactylus (Eschscholtz, 1823)" in the paper.

Comments 2: Usually in MDPI journals the reference to a figure in the text is written in full, not Fig.

Authors response: Thanks for the comment. We have made the necessary revisions throughout the text, changing "Fig." to "Figure".

Comments 3: No DOI of articles in the reference list

Authors response: Thanks for the comment. We have added the DOI after all the articles that have a DOI.

Comments 4: Please note that equipment should be fully cited with the model, company, city and country of manufacture. (For example, lines 182, 198.)

Authors response: Thank you for your comment. We have already added the complete model, company, city and country of manufacture after the device name, as described in lines 268, 291.

Comments 5: Line 244. Reference Love et al., 2014 - should be replaced by a serial number. In addition, it seems that this reference is not in the reference list.

Authors response: Thanks for the comment. We have already converted it into digital format, as described in line 354, and it has also been added to our reference list. We sincerely apologize for this oversight.

Comments 6: It is not clear why it is necessary to highlight chapter 6 Patents in a separate section.

Authors response: Thanks for the comment. We still kept this paper because it describedd the optimal temperature range for the growth of red claw crayfish (Cherax quadricarinatus). 

Comments 7: Simple summary: In my opinion, the phrase “development of its industry” is too generalized in this context. It is more appropriate to specify the need for research for aquaculture.

Authors response:  Thanks for the comment. We have made revisions to the "simple summary" and have further specified the requirements of the research, as described in lines 39-40.

Comments 8: Introduction:lines 61-63. It is not clear why the qualifier “especially for aquatic animals” is used if it says aquatic organisms. Who do you mean in the first and second cases?

Authors response: Thanks for the comment. We have revised this part. The point was not clearly expressed in the paper. We have made modifications to the entire introduction. What we wanted to convey is that, especially for poikilothermic animals like crustaceans, temperature is a very important factor, as described in lines 141-144.

Comments 9: Introduction:Lines 73-76. This sentence needs to be rephrased The verb “Rely” is misleading.

Authors response: Thanks for the comment. We have rephrased this sentence to eliminate any ambiguity, as described in lines 163-165.

Comments 10: Introduction:In general for this section, it seems to me more logical to described the species being studied first, and then move on to the relevance of the study to described the species being studied.

Authors response: Thanks for the comment. We agree with the reviewer's opinion. The original introduction section was indeed lacking in coherence. It would be more appropriate to introduce the species of narrow-clawed crayfish Pontastacus leptodactylus first. We have already revised the sequence of description in the entire introduction.

Comments 11: Introduction:In addition, it would be useful in this section to justify the advantages of using hemolymph cells and hepatopancreas tissues as “targets” over other organs and cells.

Authors response: Thanks for your constructive comment. We agree with the reviewer's point of view. In the introduction section, lines 174-178 and 192-200, we have added the advantages of using hemolymph and hepatopancreas.

Comments 12: Methodology:lines 157-160. What number of days the individuals were acclimated before the experiment. What were the hydrochemical parameters of the water?

Authors response: Thanks for the comment. The crayfish were acclimated to the laboratory conditions (temperature: 15 ± 1℃; dissolved oxygen: > 7.5 mg/L; pH: 7.8 ± 0.4; total ammonia nitrogen < 0.1 mg/L; NO2-N < 0.1 mg/L) for 2 weeks. We have provided additional information on this part in the Experimental animals and rearing conditions section, as described in lines 232-238.

Comments 13: Methodology:Lines 163-165. Please clarify: if there were 4 groups of 4 replicates, and 20 crayfish in each group, the total number of crayfish is 320 individuals, while line 159 indicates 200. Or were there 6 tanks with 200 crayfish each? Then it is not clear why so many individuals were acclimated.

Authors response: Thanks for the comment. What we need to explain is that a total of 1200 shrimps were transported from Xinjiang. First, they were placed in 6 water tanks, with 200 crayfish in each tank for temporary rearing. Then, 320 of the healthy crayfish were selected for the thermal stress experiment. There were four temperature treatment groups in total, with four replicates in each temperature group, and 20 crayfish in each group. (n = 4 replicates, 20 individuals/replicate, total 80 crayfish/treatment)

Comments 14: Methodology:Then lines 168-170. If 6 individuals were randomly selected at each time point from 4 replicates, it appears that 1 or 2 individuals were taken from each replicate. What does this have to do with it?

Authors response: We sincerely appreciate the reviewer’s attention to methodological detail. Regarding the sampling strategy: a total of six individuals randomly selected from across all four replicate tanks to ensure representative sampling of the treatment effect.

Comments 15: Results:Lines 260-262 Two sentences contradict each other. The first sentence can be removed without loss of meaning. Why is the significance level 0.05 in the text and 0.0001 in Figure 1?

Authors response: Thanks for the comment. These two sentences do have a contradiction issue. We have deleted the first sentence, as described in line 379-380. We have unified the significant differences in the text and figure to 0.05. Thank you again for your comment.

Comments 16: Figures 3 and 4. Why are the temperature points labeled T15 (20...) in the text and G15 in the figure? Indicate what the letters a,b,c stand for. Why is it stated (n = 3 replicates) when the methods say 4 replicates.

Authors response: Thanks for the comment. When the groups were set up, T was the abbreviation for "temperature", so in our text we used T. In the figure, G15 means "Group 15". We have unified the groups in the text and figure, using T15(20...) instead. Letters a, b, and c above the bars indicate that there are statistically significant differences between the groups (P < 0.05). We have marked this on each chart with letters below, in lines 451-453 and 492-494. For "n = 3 replicates and the methods say 4 replicates". Here, it's our fault for not being clear enough. At each sampling point and each temperature treatment group, we collect 6 crayfish samples in 4 parallel groups. During the sample testing process, there are 6 biological replicates and 3 technical replicates. We have already made modifications to the lines below the figure and in the methods to avoid ambiguity.

Round 2

Reviewer 1 Report

Comments and Suggestions for Authors

Accept. I suggest avoiding question-based titles. "Physiological Responses and Histopathological Changes in Narrow-Clawed Crayfish (Pontastacus leptodactylus) Under Acute Thermal Stress"

Author Response

Comments 1: Accept. I suggest avoiding question-based titles. "Physiological Responses and Histopathological Changes in Narrow-Clawed Crayfish (Pontastacus leptodactylus) Under Acute Thermal Stress"

Authors response 1: Thank you for your suggestion. Initially, our title was indeed in the declarative form. Later, one of the reviewers suggested that a question-form title would be more attractive, and he believed that no experimental study had confirmed results with 100% accuracy. However, in most cases, the title remained in the declarative form. The declarative sentences are more concise and able to directly summarize the core of the research. Readers can quickly determine the content of the paper just by looking at the title. We still preferred the declarative form, and we have revised it back. The new title is now: "Physiological Responses and Histopathological Changes in Narrow-Clawed Crayfish (Pontastacus leptodactylus) Under Acute Thermal Stress", as described in lines 2-4. Thank you again for your valuable input.

Reviewer 3 Report

Comments and Suggestions for Authors

The revised manuscript is OK but the answers provided by the authors seem very circle-like arguments and to this reviewer it seem as if whatever a reviewer would comment te author would find an answer. The revision was extremely rapid and hence one winder how much effort they have put in revising some of the point I made. Therefore there is really no need to try to suggest comments to improve this manuscript. 

Author Response

Comments 1: The revised manuscript is OK but the answers provided by the authors seem very circle-like arguments and to this reviewer it seem as if whatever a reviewer would comment te author would find an answer. The revision was extremely rapid and hence one winder how much effort they have put in revising some of the point I made. Therefore there is really no need to try to suggest comments to improve this manuscript.

Authors response: Thank you for your comments. We sincerely appreciate your efforts in reviewing our work and providing constructive comments to improve its quality. We fully understand your concern regarding the perceived rapid revision. Please allow us to clarify that we carefully considered each of your previous comments and made substantial revisions where necessary. These revisions included addressing issues such as centrifuge speed and the naming of hepatopancreatic cells, among others. We have comprehensively considered your suggestions. The journal required us to make revisions within ten days. We spent 9 days discussing and making the necessary changes, and submitted the final version on the last day. Once again, we appreciate your time and critical evaluation, which has undoubtedly strengthened our work.

Reviewer 4 Report

Comments and Suggestions for Authors

I have no new comments on the article

Author Response

Comments 1: I have no new comments on the article.

Authors response

Thank you for your time and valuable feedback throughout the review process. We sincerely appreciate your constructive comments, which have helped improve our manuscript. We are pleased to hear that you have no further comments, and we thank you once again for your contribution to enhancing the quality of our work.